# Polar Lipids in Starch-Rich Commodities to be Analyzed with LC-MS-Based Metabolomics—Optimization of Ionization Parameters and High-Throughput Extraction Protocols

**DOI:** 10.3390/metabo9080167

**Published:** 2019-08-12

**Authors:** Christin Claassen, Jürgen Kuballa, Sascha Rohn

**Affiliations:** 1GALAB Laboratories GmbH, Research and Development, Am Schleusengraben 7, 21029 Hamburg, Germany; 2Hamburg School of Food Science, Institute of Food Chemistry, University of Hamburg, Grindelallee 117, 20146 Hamburg, Germany

**Keywords:** metabolomics, starch removal, extraction, lipids, ESI ionization parameters, UPLC-IMS-QToF, potato, *Solanum tuberosum* L.

## Abstract

Metabolomics-based approaches are still receiving growing attention with regard to food authenticity testing. Such studies require enormous sample numbers with negligible experimental or analytical variations to obtain statistically reliable results. In this context, an extraction protocol in line with optimized ionization parameters was developed in consideration of potential starch-derived matrix effects focusing on the polar lipids of potatoes. Therefore, well-known extractions (Bligh and Dyer, Folch, Matyash, and a *n*-hexane-based procedure) were compared in a non-targeted and a targeted approach regarding the extractability of their lipids such as phosphatidylcholines, phosphatidylethanolamines, galacto- and glucocerebrosides, di- and triglycerides, and acylated steryl glucosides. The selected Folch method was also scrutinized in view of its ability to remove the matrix’s starch and consequently improved by substituting trichlormethane with ethyl acetate as a “greener” Folch approach. Moreover, the challenge of starch-derived contamination and imminent ion suppression in the electrospray ionization source (ESI) was addressed by an optimization of ionization parameters varying desolvation settings, removing injection peaks, and increasing the angles and distances of the ESI-device. Long-term stability tests over five days were performed successfully with a combination of appropriate extraction and decreased desolvation settings during ionization. In conclusion, the present methodology provided the basis for on-going large-scale metabolomic studies with respect to the botanical origin of potatoes using UPLC-IMS-QToF (ultra-high performance liquid chromatography ion mobility spectroscopy quadrupole-time of flight mass spectrometer).

## 1. Introduction

The plant metabolome, a complement of cellular metabolites, mirrors cell activities at a functional level. The comprehension of metabolic compositions can thereby be considered as a tool in modeling intra- and intercellular biological pathways [1]. To observe characteristic patterns, metabolic fingerprinting, as a non-targeted approach which detects as many metabolites as possible without necessarily identifying or quantifying them, is often conducted [2]. An appropriate statistical evaluation of metabolomic experiments requires data sets with enormous sample numbers. Therefore, any experimental and instrumental variations must be reduced to a minimum in order to focus on the biological information of importance [3]. 

With regard to the determination of the botanical and geographical origin as a matter of food authenticity testing, a sample set of potato tubers will be investigated in a metabolomics-based approach using LC-ESI-MS (liquid chromatography electrospray ionization mass spectrometer). In the experimental design, the composition of the matrix must be considered. According to the literature, potato tubers, used here as a model food item, consist of water (77.8%), proteins (2.0%), fibers (2.0%), minerals (1.0%), lipids (0.1%), and especially starch (14.1%) [4]. As electrospray ionization is characterized as rather insufficient for non-polar metabolites with an increased susceptibility to matrix effects [2], the adaptation of ionization parameters to the investigated matrix is inevitable. Effects of matrices with a high starch and low fat content such as maize and other cereals, resulting in the enhancement or suppression of an analyte´s response using LC-ESI-MS, were already reported in the context of mycotoxin and pesticide analysis [5,6]. Thereby, two general approaches were described to handle matrix effects: changes in the ionization methods and/or the improvement of the sample treatment applied [6]. 

Beside ionization parameters, extraction procedures must be adopted to the present matrix, especially in approaches based on metabolic fingerprinting, because of their sample clean-up steps, which are commonly reduced to a minimum. This methodology presents a serious risk of resulting in the suppression or enhancement of a metabolite´s response during the ionization process, especially in long-term measurements [5,6]. Plant matrices include many chemically diverse metabolites of varying concentration, whereby a single procedure is not capable of extracting all metabolites. The solubility of metabolites is determined by intermolecular interaction between solvent and solute substances, sometimes summarized as the “like dissolves like” rule [7]. As a result, the extraction procedure is crucial for the subsequent detection of naturally occurring metabolites in the extracted matrix [8]. Therefore, a workflow has to be established that separates the major constituents of potato tubers, while substances of interest are enriched. In this context, five biphasic extraction systems, some of which were recently tested for matrices with low fat and high water contents, such as asparagus in a metabolomic approach [9], were compared regarding their number of detected mass features and their sum of all peak areas and heights in a non-targeted approach. Furthermore, relative peak areas of various galacto- and glucocerebrosides, phosphatidylcholines and -ethanolamines, di- and triglycerides, acylated steryl glucosides, and further potato metabolites, representing a wide range of lipids, were analyzed in a targeted approach. The present study focused on lipids instead of other higher-concentrated components because this minor fraction has been less investigated to date than the major ones. As Arvanitoyannis et al. already reviewed in 2008 [10], many approaches deal with the determination of the moisture, starch content, sugars, fibers, elements, and proteins in potato tubers [11,12,13,14].

Scientific publications with the main subject of authenticity testing in plant-derived matrices are increasingly being published [15,16,17], but there is no comprehensive study that has dealt with the analytical challenges of starch-rich commodities, particularly potatoes and grains, in comprehensive LC-MS-based metabolomic experiments. Previous investigations on potato lipids were conducted using gas chromatographic analysis and gravimetric methods for the determination of diverse classes and subclasses of lipids as sum parameters in potato tubers and potato products [18,19,20]. Furthermore, targeted studies focusing on glycolipids and phospholipids in genetically-modified potatoes using normal-phase liquid chromatography were also introduced [21]. 

The aim of the present LC-MS-based metabolome study was to develop an extraction approach for starch-rich commodities (in this case, potatoes as a model food) with a special focus on enriching polar lipids to a large extent. Simultaneously, starch as a substantial part of such matrices should be entirely removed to prevent the subsequently utilized mass spectrometer from starch-derived contamination resulting in severe analytical problems. Furthermore, this imminent problem of ion suppression was addressed by optimizing ionization parameters. 

## 2. Results

### 2.1. Comparison of Extraction Procedures

Four well-known extraction procedures that demonstrate good extraction capacities of lipophilic substances, in particular procedures according to Bligh and Dyer [22], Folch [23], Matyash [24], as well as an *n*-hexane-based approach [25], were applied to the starch-rich commodity potato tubers. For the comparison of extraction yields, dimensions were reduced, and varietal differences within the sample set were therefore initially erased from evaluation by including only the respective quality control (QC) sample for each extraction approach. As all four QC samples were based on the same sample material, differences in the obtained total ion chromatograms (TICs) and peak areas could solely arise from the metabolite-dependent extractability of the selected method. The long-term extract stability was not investigated as it had been proven for the present extraction procedures already [9].

As the present methodology was developed in terms of its suitability for metabolic fingerprinting, the evaluation focused on non-targeted characteristics such as the total number of detected mass features, the sum of peak areas, and peak heights. Mean values and standard deviations were calculated from the QC samples’ technical replicates in order to investigate instrumental variations. 

#### 2.1.1. Non-Targeted Evaluation

While the trichlormethane (TCM)-based extracts revealed the highest observed numbers of detected mass features, with 32,194 compounds (Bligh and Dyer [22]) and 27,054 compounds (Folch [23]), pooled QC samples according to Matyash (24,559 masses) [24] and Reis using *n*-hexane (19,529 masses) [25] resulted in lower numbers (Figure 1). 

Regarding the sums of all peak areas and heights, TCM- and tert-butylmethylether-based (TBME) extracts exhibited similar values of 1.00 × 10^9^ counts (peak areas) and 1.35 × 10^8^ counts (peak heights) [22,23,24], whereas only 52.32% of these peak areas and 55.52% of the peak heights could be observed in the *n*-hexane-based QC sample [25]. Underlying total ion chromatograms were summarized in Appendix A. Due to the comparatively low number of detected features with low peak intensities, *n*-hexane extractions were not suitable for the collection of the potential lipophilic biomarkers that are typically present in potato tubers with low concentrations. To gain further insights into a procedure’s extractability, a targeted evaluation with regard to multiple lipid classes and subclasses was carried out.

#### 2.1.2. Targeted Evaluation

Relative peak areas of target substances were compared across the four mentioned extraction procedures. All calculated values were visualized in Appendix A. Because most target substances were not commercially available as pure standards for quantitative purposes, the determination of analytical performance characteristics such as the limit of detection or quantification was not executed.

With regard to galactocerebrosides (GalCer), glucocerebrosides (GlcCer), phosphatidylcholines (PC), and phosphatidylethanolamines (PE), the procedures according to Bligh and Dyer [22], Folch [23], and Matyash [24] exhibited similar relative peak areas, primarily ranging from 80 to 100%. The analysis of *n*-hexane based extracts [25] revealed relative peak areas of 42–53% for GalCer, while related GlcCer, PC and PE resulted in rather poor mean values. Instead, *n*-hexane extracts were repeatedly capable of achieving relative peak areas of 100% for diglycerides (DG), triglycerides (TG), and acylated steryl glucosides (ASG), being the most non-polar metabolites under investigation, which was hardly surprising due to the non-polar properties of *n*-hexane. Procedures according to Folch [23], Bligh and Dyer [22], and Matyash [24] also obtained good extraction rates concerning DG, ASG, and TG. With the increasing length of the fatty acid chain and therefore with decreasing polarity, the extraction yields of ASG accomplished by the TCM- and TBME-based extracts stagnated from more than 95% (16:0-glc-stigmasterol; glucose (glc)) to 62–80% (24:0-glc-sitosterol), while the values of the more non-polar *n*-hexane extracts [25] remained at 100%. With regard to polar potato metabolites such as α-solanine and α-chaconine, only TCM-based extraction methods, especially the Folch method [23], exhibited some extraction capacities, while the presence of those alkaloids was negligible in *n*-hexane and TBME extracts. Due to varying ratios of the organic phase to the polar phase, the Folch method [23] demonstrated significantly higher extraction capacities for polar metabolites, while the Bligh and Dyer procedure [22] resulted in slightly higher extraction rates for more non-polar metabolites of the lipid profile. Regarding semi-polar metabolites such as cholecalciferol, TBME extraction revealed the best extraction results. 

The extraction of non-polar metabolites (TG, ASG, DG) was feasible with all investigated biphasic systems. In contrast, the enrichment of more polar lipids (GalCer, GlcCer, PC, PE) was impossible within applications using *n*-hexane. For this reason, this approach was not continued. Due to its absent extraction ability concerning polar metabolites and ether-derived pressure problems during tube shaking resulting in additional steps to release pressure from surrounded tubes, the TBME-based approach was also not suitable for prospective high-throughput procedures. The extraction method according to Folch [23] was chosen for future lipid investigations in preference to that of Bligh and Dyer [22] because of the significantly higher extraction rates of polar metabolites leading to a broad coverage of different polarities and therefore a wider compilation of the lipid profile, which is an essential requirement in the field of metabolomics [8].

Apart from the comparison of extraction yields and procedures, the sample set was evaluated with regard to varietal differences. Subsequently, three extraction replicates of each variety, analyzed twice per extraction procedure, were considered, indicating significant varietal difference across all four investigated approaches. This varietal interrelation regardless of the evaluated procedure was summarized exemplarily for GalCer (23:0(OH)/18:1) in Figure 2. 

#### 2.1.3. Upscaling to Large Metabolomic Datasets

For further investigations of varietal differences as a matter of authenticity testing towards botanical origins of food, a large-scale study with authentic samples was carried out. For this purpose, 98 authentic samples were extracted with the selected Folch methodology [23] and, moreover, a QC sample including aliquots of all authentic samples was prepared. These TCM-based extracts were analyzed over five days using a desolvation temperature of 823.15 K and a desolvation gas flow of 1000 L/h (N_2_), as similar conditions had been used for reliable lipid profiling previously [26]. During the sequence, the QC sample was repetitively measured every five potato samples in order to control the system stability on the basis of TICs and peak intensity courses. The TICs of the first five QC sample injections were compared to those of the last five (injections 240–260) in Figure 3, resulting in severe losses of signal intensities and therefore in irreproducible measurements. 

Within a targeted approach, the peak intensity course of the analyzed sequence was illustrated for every second measured QC sample in Appendix A using DG (18:1/18:1) as an example. The assessment of peak intensity courses also revealed significant losses of signal intensities after an analysis duration of two days (injection number 100). This effect was presumably caused by ion suppression due to the starch-derived contamination of the ionization device that could also be observed in the form of solid residues in the instrument´s source. As an appropriate normalization was a mandatory step of data pretreatment for all metabolomics-based approaches, unwanted experimental and instrumental variations within a data set had to be eliminated [3]. Therefore, Folch extraction as well as ionization parameters were improved with regard to their ability to remove the commodity´s starch content. The aim of these enhancements was to obtain TICs without any signal intensity losses as well as a constant minimum relative peak area of 80% in QC samples (targeted) over at least five days. This limit was estimated to be a feasible value to enable a suitable normalization and the subsequent determination of biological differences prior to instrumental variations. 

At first, the described QC sample as a mirror of the lipophilic potato metabolome was deployed for the comparison of ionization parameters in the ESI source. Relevant settings were tested to address the imminent problem of ion suppression in the long-term measurements which are essential in the area of metabolomics and studies based on large sample sizes. 

### 2.2. Comparison of Ionization Parameters Regarding the Removal of Starch Residues

In order to compare the effects of analysis parameters appropriately, the TCM-based QC sample was repetitively analyzed 130 times within 2.5 days, intermitted by a blank sample every fifth measurement. For stability tests, a duration of 2.5 days was chosen, because two days turned out to be the critical point in the preceding experiment. In Figure 4, the TICs of the first five injections were compared to those of the last five (injections 126–130). 

As previously reported, the intensity loss at an analysis time of 48 h using the initial desolvation temperature of 823.15 K and a desolvation gas flow of 1000 L/h (N_2_) was confirmed with this repetition (Figure 4A). Analytical variations using these settings were also demonstrated in the context of the decreasing number of mass features from an initial 25,030 to 22,076 components, accompanied by enormous standard deviations within the last five injections, with the presumption that this number will continue dropping down in subsequent injections. Moreover, the sum of all peak areas decreased to 87.78% of the initial value (Appendix A). 

Furthermore, the QC sample was analyzed for another 2.5 days by diverting the first 1.5 min of the chromatographic run into waste instead of the ionization device. This approach was taken to remove mono- and polysaccharides deriving from the cleavage of starch as a possible method to avoid ion suppression within the instrument´s source. Compared to the first settings, the removal of the injection peak led to similar start intensities (maximum 1.50 × 10^7^) and a reduced intensity decrease over time (Figure 4B). The number of detected features remained at a constant level of 25031 masses during the analysis time, whereas the sum of peak areas also showed no significant decrease (with an end value of 96.21% of the initial peak areas) (Appendix A). As an adverse effect of this method, very polar metabolites such as α-solanine and α-chaconine were removed from the metabolomic investigation. 

In addition, the angle position of the electrospray ionization-probe (ESI) was changed from 7.0 as before to 8.5, resulting in an increased angle and distance between the probe and the ion entrance of the detector. As expected, the intensity dropped down to an initial maximum peak height of 1.00 × 10^7^ counts, proceeding to an end value of 8.00 × 10^6^ counts (Figure 4C). Beside the intensity decline, broad unresolved peaks were observed that have to be mentioned as an additional disadvantage of this approach. While the number of detected features remained static at 25031 metabolites in accordance with the sum of all peak areas (with an end value of 96.10% of the initial value), the sum of peak heights and its standard deviation revealed severe analytical variations arising from unstable conditions during electrospray ionization (Appendix A). Therefore, the sum of peak areas remained stable, and simultaneously, the sum of peak heights decreased. This context led to the assumption that already poorly-shaped peaks became increasingly broader along with decreasing heights in the meantime.

As a consequence, the application of all three ionization methods was not acceptable for high-throughput investigations of starch-rich commodities. 

Assuming that starch and starch-derived contaminants were less ionizable with a reduced desolvation temperature of 703.15 K and a desolvation gas flow of 800 L/h (N_2_), a fourth analysis of the QC sample was started with these settings. The evaluation revealed lower start peak heights (max. 9.00 × 10^6^ counts) than the previously reported approaches. Contrary to all other parameters, the initial and end peak heights corresponded to each other, exhibiting almost no signal decrease (Figure 4D). Apart from TICs, the evaluation of all detected masses revealed the highest observed numbers of 27223 and 27439 components in accordance with a static sum of all peak areas (with an end value of 96.66% of the initial sum). Also, the sum of peak heights stayed stable at 100% (Appendix A). 

Regarding the 24 metabolites described in the targeted approach, the relative peak areas and standard deviations of the initial five injections were related to those of the last five injections. The intensity courses of all investigated ionization parameters were exemplarily illustrated for DG (18:1/18:1) in Appendix A. The evaluation of this single target substance led to conclusions in line with those of the non-targeted evaluation.

Due to stable peak courses and negligible analytical variations over 2.5 days, decreased desolvation parameters were set in the future analysis method for starch-rich commodities, although the intensities of most metabolites were generally lower than those of former parameters (Appendix A).

### 2.3. Enhancement of the Extraction Method According to Folch

Due to experimental variations as well as environmental and health concerns, Folch extraction was improved by the substitution of TCM with the less-hazardous ethyl acetate (EtOAc) [27]. The Folch strategy based on a two-step method divided into a first solid/liquid and a subsequent liquid/liquid extraction resulting in the partition of hydrophobic lipids to the EtOAc-rich ternary fluid phase and the accumulation of unwanted polar substances in the water-rich phase was retained in this “greener” substitution approach related to Folch [27]. Besides having polarity properties similar to TCM, EtOAc demonstrated appropriate volatility and viscosity and the ability to form a two-phase system with water [27]. Consequently, the same authentic sample set that had previously been extracted with TCM [23] was also extracted according to the “green” Folch approach [27]. A QC sample was prepared analogously. 

In order to compare the extraction capacities of the traditional Folch method [23] to those of the “green” Folch method [27], the respective QC samples were analyzed in ten-fold replicates applying the increased initial desolvation settings. 

#### 2.3.1. Non-Targeted Evaluation

The EtOAc-based QC sample [27] revealed a higher number of detected mass features (36668 compounds) than the pooled QC sample according to Folch [23], which contained only 23428 masses (Figure 5). 

While the sum of all peak areas in TCM-based extracts exceeded that of the “green” Folch approach (1.06 × 10^9^ counts vs. 7.13 × 10^8^), the sum of all peak heights was observed to be slightly higher in EtOAc-based extracts than that from the Folch extracts (8.92 × 10^7^ vs. 9.57 × 10^7^). Consequently, the EtOAc-based extract led to more detected mass features exhibiting narrow peaks with smaller peak heights, whereas the TCM-based extraction resulted in less detected features with broader peaks and increased peak heights. Underlying total ion chromatograms are summarized in Appendix A. Furthermore, the lower number of detected features in line with a higher sum of peak areas might indicate that fewer metabolites could be extracted with TCM, but the ionization of these substances was more efficient.

For metabolic fingerprinting, the collection of as many metabolites as possible was essential [2]. Therefore, the “green” Folch approach seemed to be more suitable for further non-targeted metabolomic studies.

#### 2.3.2. Targeted Evaluation

For both procedures, the mean values and standard deviations of the considered metabolites are summarized in Appendix A.

Extraction rates of GalCer and GlcCer were significantly higher in EtOAc extracts (100%) [27] than in TCM-based ones (30% and 61% in average) [23]. Similar relations were observed within the group of PC (25–90%), PE (50–70%), and further potato metabolites (51–87%). In contrast, EtOAc-based extracts exhibited significantly decreased extraction rates of ASG, DG, and TG (29%, 40% and 75% in average).

#### 2.3.3. Detection of Starch Residues

Furthermore, starch was qualitatively detected in the QC samples of Folch [23] and “green” Folch [27] using Lugol´s solution, which could already detect a few micrograms of amylose and amylopectin per milliliter as a component of starch [28]. As a result, starch residues were determined in the TCM-based extracts accumulating at the phase transition along with proteins, whereas EtOAc extracts no longer revealed starch or other matrix residues (Figure 6). 

Besides the varying character of the two solvents, the main reason for the complete removal of starch from the EtOAc-based extracts was the enhanced effectiveness of centrifugation. While the entire starch was centrifuged down to the bottom, the upper organic EtOAc phase was easily transferred to other vials for analysis.

In summary, high-throughput measurements of the starch-rich commodity potato were carried out by combining the EtOAc-based extraction protocol with decreased ionization parameters in the analysis method. 

### 2.4. Sustainability of Signal Intensities and Analysis of Authentic Potato Samples

The botanical origin of potatoes, as a major question of authenticity, was investigated with two studies. An approach based on TCM extracts according to Folch [23] was conducted, applying increased desolvation settings of 823.15 K and 1000 L/h (N_2_), whereby the ionization device had to be cleaned and recalibrated every second day of analysis to maintain instrument performance, resulting in stable TICs with no observed signal intensity drop-downs. Additionally, relative peak areas above the required level of 80% over an analysis duration of five days were achieved. The second study was carried out by analyzing EtOAc-based extracts [27] at a desolvation temperature of 703.15 K and a gas flow of 800 L/h (N_2_) over five days without any cleaning steps in-between. The evaluation of the QC samples’ TIC (Figure 7) as well as the targeted analysis of selected metabolites revealed consistent ion intensities over five days which were demonstrated with DG (18:1/18:1) as an example, as shown in Appendix A. 

As the successful removal of starch and other matrix components was accompanied with more reproducible static signal intensities without instrumental contamination or ion suppression, the correlation between starch and matrix effects was evident.

As experimental and instrumental variations were eliminated, the two obtained data sets were evaluated with the help of multivariate analysis after respective data pretreatment. For the distinction of the potato varieties Belana, Gunda, Wega, and Queen Anne, a principal component analysis (PCA) was conducted for each data set by including only a selection of highly discriminant metabolites as another statistical approach [29]. For this purpose, metabolites were chosen whose abundance complied with the following conditions: a minimum two-fold change between the varietal groups, a maximum standard deviation of 30% within the groups, a one-way analysis of variance (ANOVA) with a probability (p) of more than 95% highlighting the difference between the varietal groups, a predictive power above 80% when increasing the number of samples, and a q-value describing the chance of false positives below 0.1% [30]. 

The resulting peak lists included 19 (in TCM) and 112 (in EtOAc) masses consisting of potential varietal markers. PCAs based on these peak lists are shown in Figure 8 for each data set.

In conclusion, the application of EtOAc combined with decreased desolvation settings led to the clearer segregation of the four varietal clusters within the respective PCA than the TCM-based method with increased settings. Furthermore, the “green” Folch approach [27] provided a larger number of highly significant substances, whereas the traditional Folch study [23] resulted in a smaller number and a separation of only two clusters within the PCA: Belana and Gunda in one, and Wega and Queen Anne together in a second cluster. Presumably, these two groups could also be distinguished in TCM extracts, because their composition primarily differed in terms of ASG, DG, and TG, which exhibited better extraction rates in TCM than in EtOAc. The observed segregation of Belana from Gunda as well as Wega from Queen Anne into four varietal clusters in EtOAc extracts led to the assumption that these two varietal pairs could solely be differentiated considering the more polar GalCer, GlcCer, PC, and PE that were highly concentrated in EtOAc phases. The metabolomic investigation of varietal variations with respect to further metabolic influences and the identification of marker substances will be carried out in an additional study [30]. 

## 3. Discussion

A recent study investigated lipophilic extraction procedures for plant matrices with low fat, low protein, and high water contents such as *Asparagus officinalis*, determining the Bligh and Dyer method as the most suitable method for such matrices [9]. In contrast to this recent study, the comparison of four well-known extraction procedures for the enrichment of lipids in potato tubers resulted in the selection of the Folch method, which exhibited a comparatively high number of detected features with high sums of peak areas and heights. Moreover, better extractabilities over a broad range of metabolites from potato-specific α-solanine and α-chaconine to triglycerides were obtained with the Folch rather than with the Bligh and Dyer method. With regard to the starch-rich commodity of potato tubers, the Folch method was more suitable for the extraction of as many metabolites of divergent polarities as possible than the related method according to Bligh and Dyer. Consequently, the Folch procedure provided the basis for the first ensuing long-term measurements.

Previous studies using LC-ESI-MS revealed enormous analytical interferences through enhanced or suppressed analyte responses in the context of matrices with high starch and low fat contents such as maize or other cereals during the targeted analysis of mycotoxins and pesticides [5,6]. Thereby, two approaches to reduce mainly starch-derived matrix effects to a minimum level were described: the improvement of the ionization methods applied or the sample treatment itself [6]. Methodologies based on metabolic fingerprinting such as that presented here were reported to have a high risk of resulting in analytical variations due to skipped or minimum sample clean-up steps. Thereby, the analysis was affected particularly during the electrospray-ionization process in long-term measurements [5,6]. 

In compliance with previous research, the presented approach using the combination of the Folch extraction in line with increased desolvation settings of 823.15 K and 1000 L/h (N_2_) led to suppressed responses of the exemplarily selected metabolite DG (18:1/18:1) as well as decreasing signal intensities in the TICs of the first five injections compared to those of the last injections after an analysis duration of five days. As the effect of ion suppression was consolidated after two days of analysis, matrix effects only cause analytical variations in the long-term measurements which are essential for all metabolomics-based approaches which require large sample sets for statistical reliability. Metabolomics-based approaches such as metabolic fingerprinting are non-targeted, and thus an analysis will be carried out without the application of external or internal standards [8], whose peak areas could have been considered for the correction of slight analytical variations, as the analysis is usually conducted in a targeted way [5]. Due to missing opportunities for correction—e.g., via external or internal standards—experimental and analytical stability without any variation is inevitable in the field of metabolomics. On the basis of stable conditions, metabolomics approaches can be evaluated using multivariate data analysis to discover biological variations of interest that are not covered by potential analytical interferences [3]. Possible matrix effects should be considered especially in lipidomics and LC-ESI-MS-based studies, because the ionization of lipids is rather insufficient at any rate [2].

In this context, the underlying extraction method as well as the ionization parameters applied were optimized to achieve analytical stability for future metabolomics-based approaches such as the determination of a potato tuber´s botanical origin as a matter of food authenticity testing. As severe analytical problems were presumably derived from the matrix’s main component of starch, special attention was paid to the removal of this with the simultaneous preservation of lipid detection. 

Ionization parameters were varied in terms of an increased angle and distance of the ESI probe towards the ion entrance as well as by diverting the first 1.5 min of the chromatographic run into waste. Both approaches were not suitable for ongoing analysis because of their obvious disadvantages regarding the obtained broad and unresolved peaks or the removal of potato-specific metabolites such as α-solanine. The fourth selection of ionization parameters using 703.15 K and 800 L/h (N_2_) was applied successfully over an analysis duration of 2.5 days. Consequently, the improvement of ionization methods reduced the suppression of analyte responses during long-term measurements, as already stated in the literature [6].

As the second parameter for the reduction of matrix effects, the extraction procedure, required ongoing development, environmental and health concerns were thereby taken into account. The hazardousness of TCM has already been well described. Therefore, many modifications of TCM-based extraction procedures with the aim of substituting the toxic and carcinogenic TCM with less hazardous solvents were conducted previously [27]. This substitution within a ternary solvent composition required the selected solvent to possess similar physicochemical properties to retain the two main steps of the Folch approach: solid/liquid extraction with a subsequent liquid/liquid partitioning step [27]. Besides the reduced environmental and health risks, the selected “green” Folch approach using ethyl acetate also led to an improved separation of the matrix’s starch. The main reason for the removal of the matrix’s components was that the upper EtOAc phase was used for subsequent LC-MS analysis (in contrast to the bottom TCM phase of the Folch method). Thereby, the centrifugation was more effective, and the danger of accidentally catching some matrix residues during transfer in the analysis vial was reduced. As a result, LC-MS analysis of the upper phase (of a biphasic extraction) is generally recommended for future long-term metabolomic investigations of starch-rich commodities such as potato tubers. 

The combination of the “green” Folch approach in line with decreased ionization settings of 703.15 K and 800 L/h (N_2_) resulted in stable signal intensities in both evaluations (non-targeted and targeted) during an analysis time of 2.5 days. Therefore, an authentic sample set was extracted according to the “green” Folch method and analyzed, with these decreased settings revealing no analytical variations over a period of five days. The interpretation of the LC-MS data (with regard to the distinction of potato varieties) exhibited high numbers of potential varietal markers resulting in clear segregation of clusters in the score plot of the respective PCA, whereas the original Folch approach demonstrated fewer possible biomarkers with no clear separation of all four varietal clusters in the score plot of the PCA. 

The main finding of the presented study was that authentic sample sets of statistically reliable sample sizes could only be analyzed without any analytical variations using decreased desolvation settings during electrospray ionization in line with an improved “green” Folch extraction procedure. Consequently, these conclusions will be deployed for future LC-MS studies based on metabolic fingerprinting with regard to the botanical and geographical origin of starch-rich potato tubers or the determination of the applied cultivation method. 

## 4. Materials and Methods

### 4.1. Reagents and Chemicals

HPLC-grade water was purchased from LGC Promochem GmbH (Wesel, Germany). LC-MS-grade 2-propanol and formic acid (FA; 99 mL/100 mL) were obtained from Biosolve B.V. (Valkenswaard, Netherlands), while LC-MS-grade methanol (MeOH) and acetonitrile (ACN; >99.9 mL/100 mL) were purchased from VWR International GmbH (Darmstadt, Germany). 3,5-di-tert-butyl-4-hydroxytoluol (BHT; >99 g/100 g) was obtained from Fluka Chemie GmbH (Buchs, Switzerland), while HPLC-grade trichlormethane was purchased from Carl Roth GmbH and Co. KG (Karlsruhe, Germany). GC-FID-grade (gas chromatography-flame ionization detector) *n*-hexane, ethyl acetate, and tert-butylmethylether as well as potassium iodide and iodine (both ≥99.99 g/100 g) were obtained from Merck KGaA (Darmstadt, Germany), while potato starch was purchased from EDEKA AG + Co. KG (Hamburg, Germany). LC-MS-grade ammonium formate (>99 mL/100 mL) was obtained from Sigma-Aldrich GmbH (Taufkirchen, Germany) and the lock mass leucine enkephalin was obtained from Waters Corp. (Milford, MA, USA).

The reference standards 1-palmitoyl-2-oleoyl-3-linoleoyl-glycerol (TG (16:0/18:1/18:2)), 1,2-dioleoyl-*sn*-glycerol (DG (18:1/18:1)), cholecalciferol, α-chaconine, α-solanine, and 1-hexadecanoyl-2-octadecenoyl-*sn*-glycero-3-phosphocholine (PC (16:0/18:1)) were purchased from Sigma-Aldrich GmbH (Taufkirchen, Germany). Acylated steryl glucosides (ASG), (Phrenosin bottom-spot) galactocerebrosides (GalCer), and plant-based glucocerebrosides (GlcCer) were obtained from Matreya LLC (State College, PA, USA). The composition of standard solutions purchased from Matreya LLC was summarized in Appendix A.

### 4.2. Potato Samples

Four commercial samples belonging to the varieties Annabelle, Belana, Gunda, and Linda were purchased in 2016 from local retailers. Furthermore, 98 authentic potato samples of the varieties Belana, Gunda, Wega, and Queen Anne were collected in 2017. Sampling was carried out in different German cultivation regions considering conventional and organic production methods. The authentic sample set comprised 22 samples of Belana, 21 of Gunda, 32 of Wega, and 23 samples of Queen Anne. Each sample consisted of at least ten tubers resulting in a minimum weight of 1.1 kg according to common methods of representative sampling [31,32].

### 4.3. Sample Preparation and Extraction

The commercial tubers were washed with cold running tap water and cut with a ceramic knife into four equally sized pieces by avoiding green or infected areas. Two opposite quarters were ground with dry ice in a ratio of 1:1 using a Robot Coupe Blixer 3 bowl cutter equipped with a stainless-steel knife and a stainless-steel bowl (GEV GmbH, Bergkirchen, Germany). Afterwards, ten grams of the homogenate were transferred into a 50-mL tube (Sarstedt AG + Co. KG, Nuembrecht, Germany) and freeze-dried for 48 h using an Alpha 1–4 LDPlus freeze dryer (Martin Christ GmbH, Osterode am Harz, Germany). Authentic potato tubers were sent in directly after harvesting without any storage time. After receipt, samples were stored at 281.15 K until homogenization was carried out pursuant to the procedure for commercial samples.

To extract polar lipids, various extraction methods described by Bligh and Dyer [22], Folch [23], Reis and others using *n*-hexane [25,33,34] and Matyash [24] were conducted in a modified approach. Therefore, lyophilisates of the commercial potato tubers were utilized. Each of the four varieties was extracted in triplicate, resulting in 12 samples per extraction type. The procedures of the applied four biphasic extraction methods were summarized in Table 1. All extraction steps were thereby carried out on crushed ice.

All lyophilisates were endowed with six 5-mm-glass beads (Carl Roth GmbH + Co. KG, Karlsruhe, Germany). Following the addition of acidified organic solvents including BHT as an antioxidant, all samples were shaken using a VIBA 300-Collomix (Collomix GmbH, Gaimersheim, Germany). After further additions of organic solvents and/or water and repetitions of shaking, all two-phase extracts were centrifuged for 10 min at 3846 g and room temperature (Heraeus Multifuge X3, Thermo Scientific GmbH, Munich, Germany).

In order to normalize the quantity of lyophilisate used in proportion to the applied volume of organic solvent and to prevent chromatographic effects caused by different solvents, a calculated volume of the organic phase was evaporated to dryness under in a nitrogen current (N_2_) using a six-port mini-vap evaporator (Sigma-Aldrich Chemie GmbH, Steinheim, Germany). Consequently, 1.2 mL (Bligh and Dyer) [22] and 1.8 mL (Folch) [23] of the bottom TCM phase, 2.6 mL of the upper TBME phase (Matyash) [24] as well as 2.8 mL of the upper *n*-hexane phase [25] were evaporated to dryness. Residues were resolved in 1 mL TCM/MeOH (*v*/*v*; 1/1) and 1 mL MeOH (both containing 0.1 mL/100 mL FA) using a Vortex tube shaker (VWR International GmbH, Darmstadt, Germany) followed by a filtration using a Chromafil Xtra RC-20/25 syringe filter with a pore size of 0.22 µm (Macherey-Nagel GmbH + Co. KG, Dueren, Germany). The filtrates were used for LC-MS-analysis.

Lyophilisates of the authentic potato tubers were extracted according to Folch [23] as described in Table 1. The procedure corresponded to the one performed for commercial samples, except for the evaporation to dryness, which was skipped within the authentic sample set. Consequently, the TCM phase was filtered directly, and the filtrate was used for LC-MS-analysis. In addition, the same authentic sample set was extracted with an EtOAc-based approach, called the “green” Folch method [27]. This “green” approach was modified with regard to the alcohol applied [27]. Methanol was used instead of ethanol due to difficulties with the purity of ethanol deriving from the denaturants used. The protocol of this additional approach was also outlined in Table 1. Subsequent to the centrifugation for 15 min at 3846 g and room temperature, the upper EtOAc phase was filtered using a syringe filter. The filtrate was used for LC-MS-analysis and the determination of starch. Thus, 0.15 g iodine and 0.50 g potassium iodide were solved in 50 mL water to obtain Lugol´s solution [28]. For starch detection, 0.1 mL Lugol´s solution was added to 0.4 mL of the filtrates and 0.3 mL water in order to clarify phase separation after shaking the 2 mL micro-tubes (Sarstedt AG + Co. KG, Nuembrecht, Germany) manually. A positive control sample including 5 mg potato starch solved in 0.7 mL water as well as a negative control sample containing 0.7 mL water were prepared analogously.

### 4.4. UPLC-IMS-QToF Analysis

In addition to commercial sample extracts, blank samples which had been extracted analogously to the potato samples excluding the lyophilisates were analyzed. Standard solutions containing TG (16:0/18:1/18:2), DG (18:1/18:1), cholecalciferol, PC (16:0/18:1), ASG, GalCer, GlcCer, α-chaconine, and α-solanine were measured as well. Furthermore, four quality control samples (QC) were prepared in order to verify the system stability during the whole analysis time concerning chromatography and the qualitative and quantitative detection of compounds. For each of the four extraction procedures, aliquots of 150 µL of each of the 12 sample extracts were pooled to obtain the specific QC sample. In order to avoid the saturation of the detector and to obtain reliable results within the dynamic range of the detector, various dilutions of the QC samples were made and analyzed to monitor linear correlation. Analogously, different injection volumes were tested to evaluate linearity.

Prior to the start of the sample measurement, performed in a randomized order (due to instrumental drifts), ten acetonitrile-based blank samples and eight QC samples (duplicates of all four QC samples) were analyzed to ensure chromatographic stability. These chromatographic runs were not involved in the following data analysis. During the sequence, all QC samples were repetitively measured every five potato samples with a blank sample beforehand. Every sample extract was analyzed in duplicate. 

According to Vorkas et al. [26], non-polar extracts were analyzed with an Acquity I-Class UPLC (ultra-high performance liquid chromatography) system coupled to a Vion IMS-QToF-MS (Ion mobility spectroscopy quadrupole-time of flight mass spectrometer) equipped with an electrospray ionization source (ESI) (all three Waters Corp., Milford, MA, USA). Chromatographic separation of lipid metabolites was performed with an Acquity UPLC BEH C18 column (150 × 2.1 mm i.d., 1.7 µm) attached to an Acquity UPLC BEH C18 precolumn (5.0 × 2.1 mm i.d., 1.7 µm; both Waters Corp., Milford, MA, USA) at 328.15 K applying a flow rate of 0.3 mL/min. Water with 10 mmol/L ammonium formate and 0.1 mL/100 mL FA was used as mobile phase A, while mobile phase B consisted of 2-propanol/acetonitrile (*v*/*v*; 70/30) containing equal concentrations of buffer salt and acid. The following gradient settings were chosen: 0.0 min (30.0% A), 3.0 min (30.0% A), 18.0 min (0.0% A), 23.0 min (0.0% A), 24.0 min (30.0% A), and 27.0 min (30% A). 2 µL of each sample extract were injected. Samples were kept at 283.15 K in the autosampler.

Metabolites were detected in a positive ion mode within a mass range from 50 up to 1000 m/z using a Vion IMS-QToF-MS. The ESI probe angle position was set at 7. A source temperature of 393.15 K was applied using 50 L/h cone gas (N_2_), while desolvation was performed at 823.15 K with a desolvation gas flow of 1000 L/h (N_2_). Further parameters were set in the mass spectrometer: a capillary voltage of 3.00 kV, scan time of 0.300 s, and a low collision energy of 6 eV. Each precursor ion was fragmented by using a ramp with elevated collision energy (MS^E^) starting at 15.00 eV and ending at 50.00 eV. Lock mass correction was carried out with leucine enkephalin every 2.50 min. 

The analysis of authentic potato samples that were extracted according to Folch [23] corresponded to that performed for commercial samples. Aliquots of 50 µL of each sample extract were pooled to obtain the QC sample. The analysis of this QC sample was iterated by modifying ionization parameters in order to sustain long-term signal stability with regard to the possible starch residues remaining in sample extracts. For this purpose, the injection peak (first 1.5 min of the chromatographic run) was diverted into waste to decrease the ion exposure of the system through starch-derived mono- or polysaccharides. Moreover, the ESI probe position was altered from 7 to 8.5, inducing an increased angle and distance between the ionization device and IMS-QToF-MS. Additionally, desolvation settings were decreased to 703.15 K with a desolvation gas flow of 800 L/h (N_2_), leading to lower ionization rates of the potential starch residues.

The analysis of authentic EtOAc-based potato extracts was performed using decreased desolvation settings of 703.15 K and a gas flow of 800 L/h (N_2_). Aliquots of 50 µL of each sample extract were pooled to obtain the QC sample.

To avoid instrumental variations of interbatch measurements [29], the Folch-based QC sample [23] containing commercial samples was analyzed and compared to previous measurements in the context of chromatographic and detector performance prior to each subsequent analysis.

### 4.5. Data Analysis and Statistics

Extraction procedures and ionization parameters were compared in a non-targeted approach with regard to their number of detected mass features, the sum of all peak areas, and the sum of all peak heights that were obtained from the respective total ion chromatograms (TIC). These parameters were selected to gain insight into a method´s feasibility to cover as many metabolites as possible with appropriate peak shapes and intensities for large-scale metabolic fingerprinting.

In addition, extraction procedures and ionization parameters were compared in a targeted approach in consideration of three potato metabolites from the respective lipid class or subclass of GalCer, GlcCer, PC, PE, DG, TG, ASG, and other potato metabolites such as α-solanine and α-chaconine. The majority of compounds were identified with reference standards. The structural elucidation of additional metabolites such as phosphoethanolamines was performed considering the retention time, exact monoisotopic mass, MS^E^ fragmentation data, and collisional cross sections (CCS) as a degree of a metabolite´s spatial arrangement. For this purpose, the MS/MS spectral databases METLIN, Lipid Maps, and Pubchem were also consulted [35,36,37]. 

For an appropriate comparison of extraction protocols, the (relative) peak areas of the described lipophilic substances were determined in all analyzed extracts, taking into account the ratio of the lyophilisate to the volume of organic solvent. Consequently, the concentration of lyophilisate in the final analyzed extract would have been included in all comparisons if it had not been already considered in the experimental design. Relative peak areas were calculated by setting the peak with the highest observed peak area to 100%, assuming that this peak represents the maximum possible extraction yield. The remaining peak areas were related to this major peak. Evaluations were completed with the determination of mean values and relative standard deviations from replicates. All calculations were executed with Excel 2016 (Microsoft Corp., Redmond, WA, USA). The determination of analytical performance characteristics such as the limit of detection or quantification was not executed because most metabolites were not commercially available as pure reference standards for quantitative purposes. Furthermore, these parameters for single metabolites were of minor importance in non-targeted metabolic fingerprinting, which focuses on the detection of as many metabolites as possible [2]. 

The peak picking, retention time alignment, and normalization of the high-resolution profile data (two sets of authentic samples) were performed with the software Progenesis QI 2.3 (Nonlinear Dynamics Ltd., Newcastle upon Tyne, UK). The peak picking algorithm applied was not restricted with regard to sensitivity levels, chromatographic peak width, or retention time windows. Retention times were aligned to a pooled QC sample; meanwhile, all runs were normalized to their total ion intensity. Mass correction was conducted on the basis of the lock mass leucine enkephalin. The grouping of different adduct ions into one compound, called deconvolution, was conducted to determine a metabolite´s exact monoisotopic mass. The pretreated data were transferred to multivariate analysis. PCA was carried out in Progenesis QI to differentiate all four varieties under investigation. Within this approach, mass features were reduced from tens of thousands of detected ones to a small number of significant features using statistical tools such as the one-way analysis of variance (ANOVA), predictive power, or the probability of false positives. The calculation of the effect size F for the subsequent determination of predictive power per metabolite was based on the α-error (0.05), the total sample size per data set, the number of groups in line with their actual mean abundance values, standard deviations, and sample sizes per varietal cluster [38]. 

## 5. Conclusions

Four extraction procedures were investigated in the context of their ability to enrich polar lipophilic metabolites for metabolic fingerprinting. Due to the comparatively low number of detected features with low peak intensities, *n*-hexane extractions were not suitable for the collection of potential lipophilic biomarkers, whereas a TBME-based approach was not continued because of ether-derived pressure problems during sample extraction impeding the feasibility of high-throughput analysis. The Folch extraction method was selected for further enhancement in preference to that according to Bligh and Dyer due to its comparatively increased extractability regarding polar metabolites and it covering a broader range of metabolites, which is essential for non-targeted metabolomic studies. The application of the Folch approach in large-scale studies revealed severe matrix-derived analytical variations that required the ongoing improvement of the extraction procedure and ionization parameters. Therefore, desolvation temperatures and gas flows, the removal of injection peaks, and ESI probe angles were varied. In conclusion, decreased settings of 703.15 K and 800 L/h (N2) led to the highest number of detected peaks including static sums of all peak areas and heights over 2.5 days. Although the optimized parameters cannot be applied to other manufacturers´ instruments without slide adjustments, other researchers can conclude that decreased ionization settings are most suitable for long-term LC-MS-based lipidomic measurements. In contrast, increasing ESI probe angles will not prevent the instrument from starch-derived analytical problems. 

The improvement of the extraction procedure by substituting hazardous TCM with EtOAc also resulted in a comparatively high number of detected features with a reproducible appropriate peak shape and intensity. In a large-scale metabolomic experiment, the combination of decreased ionization settings in line with an EtOAc-based extraction was compared to the initial combination of increased ionization parameters and a TCM-based extraction method. The improved methodology revealed more significant potential biomarkers for the determination of a potato’s botanical origin as a matter of food authenticity testing than the original combination of ionization parameters and extraction procedures. In conclusion, future metabolic fingerprinting will be carried out using decreased ionization settings combined with an EtOAc-based extraction protocol. 

## Figures and Tables

**Figure 1 metabolites-09-00167-f001:**
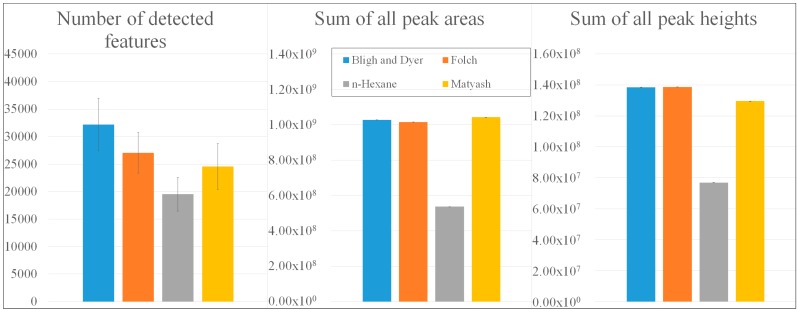
Comparison of extraction procedures according to Bligh and Dyer [22], Folch [23], Reis using *n*-hexane [25], and Matyash [24] with regard to the number of detected mass features, the sum of all peak areas, and the sum of all peak heights.

**Figure 2 metabolites-09-00167-f002:**
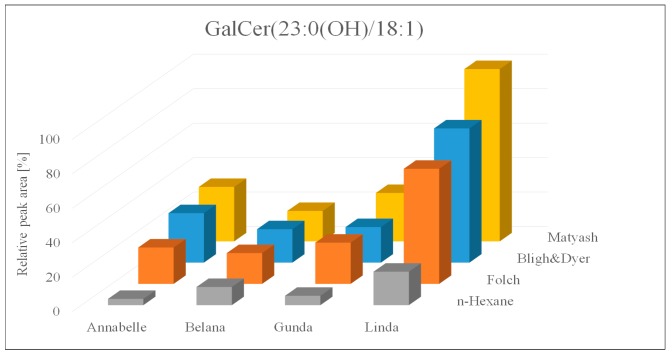
Relative peak areas of the galactocerebroside GalCer (23:0(OH)/18:1) across four extraction protocols and four different potato varieties.

**Figure 3 metabolites-09-00167-f003:**
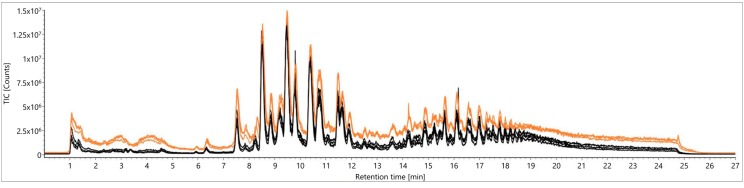
Comparison of total ion chromatograms (TICs) from a quality control (QC) sample according to Folch [23] for signal stability over five days: first five injections (orange) versus the last five (injections 240–260, black).

**Figure 4 metabolites-09-00167-f004:**
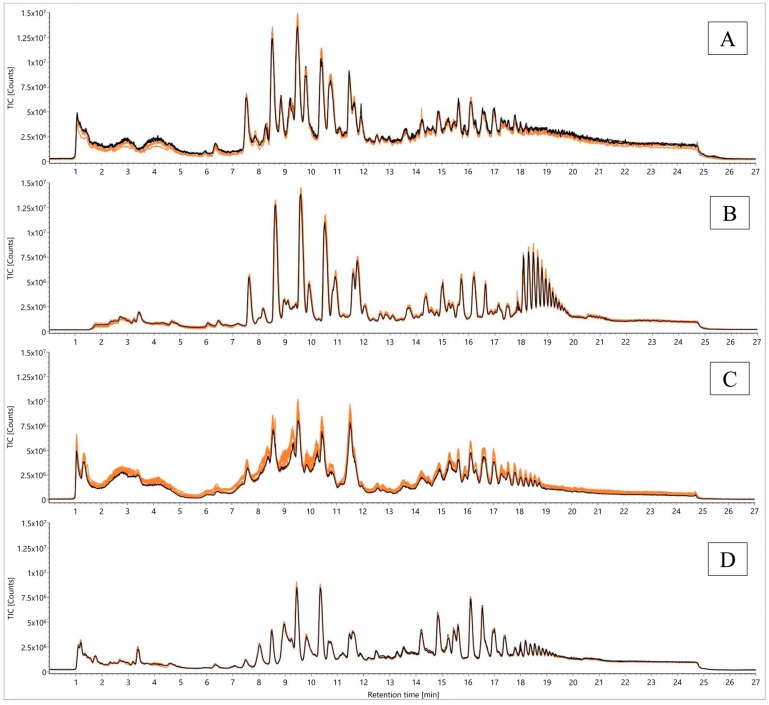
Comparison of TICs from a QC sample according to Folch [23] for signal stability over 2.5 days using different analysis parameters (desolvation temperature and gas flow, removal of injection peak, electrospray ionization source (ESI) probe angles): (**A**) 823.15 K, 1000 L/h, ESI probe angle position 7; (**B**) 823.15 K, 1000 L/h, ESI probe angle position 7, waste until 1.5 min; (**C**) 823.15 K, 1000 L/h, ESI probe angle position 8.5; (**D**) 703.15 K, 800 L/h, ESI probe angle position 7; first five injections (orange) versus the last five (injections 126–130, black).

**Figure 5 metabolites-09-00167-f005:**
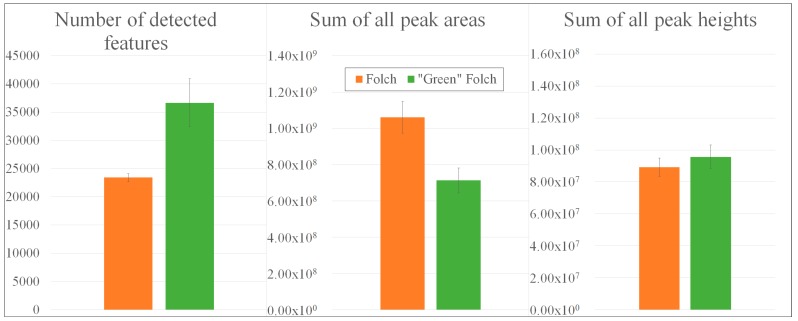
Comparison of extraction procedures according to Folch [23] versus a “green” Folch approach [27] with regard to the number of detected mass features, the sum of all peak areas, and the sum of all peak heights.

**Figure 6 metabolites-09-00167-f006:**
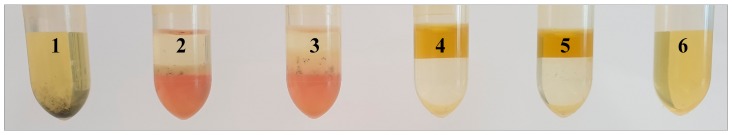
Comparison of QC samples extracted according to Folch (2 and 3) with QC samples extracted using ethyl acetate (EtOAc) (“green” Folch, 4 and 5) with regard to their starch content after centrifugation and filtration (starch detection with Lugol´s solution containing potassium iodide and iodine); 1: positive control sample, 6: negative control sample.

**Figure 7 metabolites-09-00167-f007:**
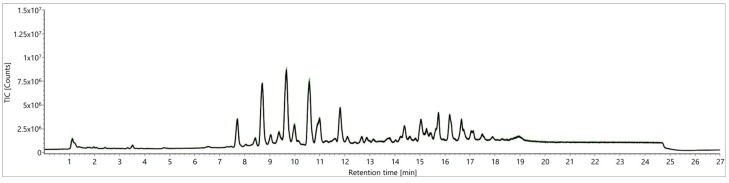
Comparison of TICs from a QC sample from a “green” Folch approach [27] for signal stability over five days: first five injections (green) versus the last five (injections 240–260, black).

**Figure 8 metabolites-09-00167-f008:**
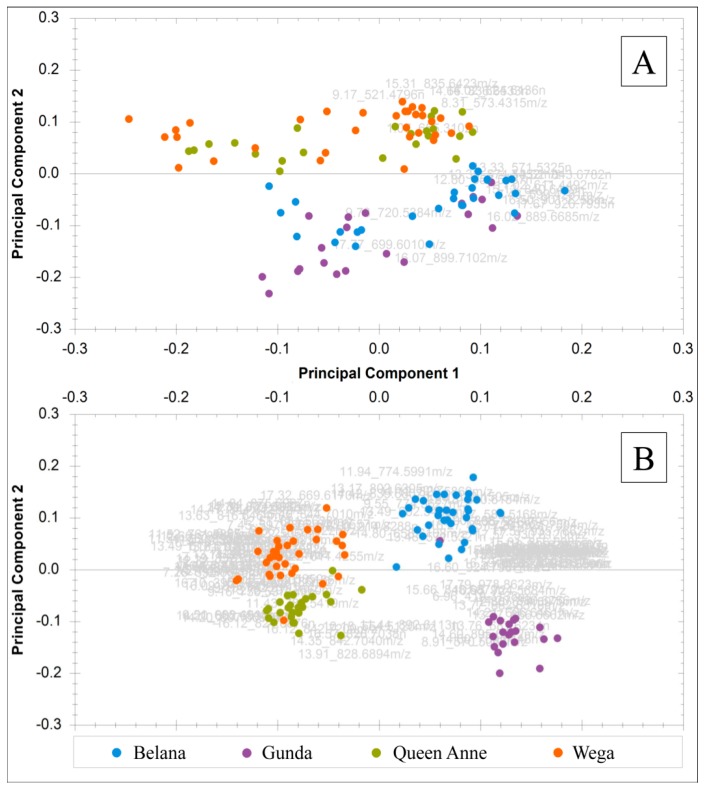
Combined principal component analysis (PCA) score and loading plots (bi-plots) of authentic potato samples displaying principal component 1 versus principal component 2: (**A**) samples extracted according to the Folch method using 19 filtered mass features with PC-1 36.07% and PC-2 23.95%; (**B**) samples extracted with EtOAc using 112 filtered mass features with PC-1 41.06% and PC-2 15.74%.

**Table 1 metabolites-09-00167-t001:** Procedures of five biphasic extraction methods to enrich polar lipids (all organic solvents contained 50 μg/mL 3,5-di-tert-butyl-4-hydroxytoluol (BHT) and 0.1% formic acid (FA); water included 0.1% FA). TCM: trichlormethane.

Bligh and Dyer [22]	Folch [23]	*n*-Hexane [25]	Matyash [24]	“Green” Folch [27]
1.0 g lyophilisat	1.0 g lyophilisat	1.0 g lyophilisat	0.5 g lyophilisat	1.0 g lyophilisat
+6 glass beads				
+5 mL TCM +10 mL MeOH	+10 mL TCM +5 mL MeOH	+25 mL *n*-hexane/2-propanol (*v*/*v*; 3/2)	+6.3 mL MeOH	+8.3 mL EtOAc/MeOH (*v*/*v*; 70/30)
5 min shaking				
+5 mL water	+4.7 mL water	+6.3 mL *n*-hexane/2-propanol (*v*/*v*; 3/2)	+7.8 mL TBME	+8.3 mL EtOAc/MeOH (*v*/*v*; 70/30)
5 min shaking				
+5 mL TCM	+7.8 mL TCM/MeOH (*v*/*v*; 2/1)	+5 mL MeOH +5 mL *n*-hexane	+7.8 mL water	+9.1 mL water +1.5 mL EtoAc
5 min shaking			Two times 1 min shaking, releasing pressure every minute	5 min shaking
+4 mL water		+5 mL water	+3.1 mL TBME	
5 min shaking		5 min shaking	4 min shaking, releasing pressure every two minutes	
			+4.7 mL water	
			3 min shaking	
Centrifugation for 10 min at 3846 g				Centrifugation for 15 min at 3846 g

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
