# Peer review of "Polar Lipids in Starch-Rich Commodities to be Analyzed with LC-MS-Based Metabolomics—Optimization of Ionization Parameters and High-Throughput Extraction Protocols"

_metabolites, 2019, doi:10.3390/metabo9080167_

Round 1

Reviewer 1 Report

Review of MS Polar lipids in starch-...

This paper is a good example of novel extraction procedure with less amount of chlorinated solvents for the analysis of plant lipids. I would publish it after minor changes, as follows.

The authors are studying polar lipid components in potatoes for the evaluation of authenticity. However, lipids are only 0.1% of potatoes. The authors should state why they studied lipids instead of other chemical markers present in higher amount, such as starch or protein components.

Line 14. ‘Such studies require enormous sample numbers without any experimental or analytical variations to obtain statistically reliable results, because...’ This sentence is dogmatic. It is impossible to get real-world samples with no variance. Please motivate this judgement.

Line 26. ‘instrumental angles and distances.’ This sentence is not understandable: are you referring to complex bidimensional chromatograms or to the geometry of the detector?

Line 35-36 ...metabolome is a complement...

Line 112-114 please cite Matyash here.

Fig. 2. How were ’relative peak areas’ calculated? Relative to what? It seems to me that in all cases their sum is much higher than 100%.

Line 169. Desolvation took place at 550 °C (823 K); what about the stability of analytes at this temperature?

Can the authors presume the concentration of the analytes of interest? At least in the order of (mg/L)?

Reviewer 2 Report

This is a thoroughly conducted study with interesting results even though it reports exclusively the development of the method that is expected to be utilized in further experiments. All in all this study warrants publications after a few issues are addressed.

1) The checks carried out encompassing parameters of the particular instrument used cannot be applied to other instruments, though data reported are of interest.

2) Did the authors checked that the detector was not saturated as intensities of 10^9 are extremely high? The same experiments with diluted samples should have been run for that.

3) Figures 1 and 5: lower number of features gives similar of higher summe of areas. This fact may indicate that some metabolites are less extracted of that they undergo more efficient ionization. The authors should take into account this feature. Analysis of diluted samples, as suggested in the point 2, may be relevant for the aim of this study.

4) Paper on mycotoxins and pesticides are referenced in regard to some measurement drawbacks, but the main aim of the study is pointed out to be lipids.

5) Experiments of enrichement of diluted extracts with pure standards would be welcome for some issue clarification

6) Extraction efficiency is expected to be directly related to the similarity of molecule and solvent polarity. The following references are recomended to be taken into cosideration: 

Reichardt C (2003) Solvents and solvent effects in organic chemistry, vol 3rd edition. Wiley, New York

Montero et al. (2013). Food Analytical Methods 6: 1497 (D.O.I.: 10.1007/s12161-013-9661-1 

Reviewer 3 Report

I have received the manuscript entitled: “Polar lipids in starch-rich commodities to be analyzed with LC-MS-based metabolomics – Optimization of ionization parameters and high-throughput extraction protocols” in order to evaluate its value to be published in Metabolites Journal.

The experiment is good planned and manuscript, written by Christin Claassen and colleagues, is well written. The aim of the described study was developing an extraction approach for starch-rich commodities (in that case potatoes). Additionally, optimizing ionization parameters was performed. The manuscript is well prepared. I really like even the graphical side of the presented results, consequence in using colors etc. I appreciate that developed methodology was tested on authentic samples. I don't feel qualified to judge about the English language and style, however, it looks like English language and style are fine.

According to my knowledge, the manuscript is good and does not contain major flawls. Maybe some small but not necessary corrections could be performed, however, in my opinion, the manuscript can be published even in present form. I think it can be interested for the readers from that research area.
